# Genomic and Epidemiological Features of COVID-19 in the Novosibirsk Region during the Beginning of the Pandemic

**DOI:** 10.3390/v14092036

**Published:** 2022-09-14

**Authors:** Natalia Palyanova, Ivan Sobolev, Alexander Alekseev, Alexandra Glushenko, Evgeniya Kazachkova, Alexander Markhaev, Yulia Kononova, Marina Gulyaeva, Lubov Adamenko, Olga Kurskaya, Yuhai Bi, Yuhua Xin, Kirill Sharshov, Alexander Shestopalov

**Affiliations:** 1Laboratory of Molecular Epidemiology and Biodiversity of Viruses, Research Institute of Virology, Federal Research Center of Fundamental and Translational Medicine, 630117 Novosibirsk, Russia; 2Department of Natural Science, Novosibirsk State University, 630090 Novosibirsk, Russia; 3CAS Key Laboratory of Pathogenic Microbiology and Immunology, Collaborative Innovation Center for Diagnosis and Treatment of Infectious Disease, Institute of Microbiology, Center for Influenza Research and Early-Warning (CASCIRE), Chinese Academy of Sciences (CAS), Beijing 100101, China; 4China General Microbiological Culture Collection Center, Institute of Microbiology, Chinese Academy of Sciences, Beijing 100101, China

**Keywords:** SARS-CoV-2, first wave, Russia, epidemiology, COVID-19, phylogeny

## Abstract

In this retrospective, single-center study, we conducted an analysis of 13,699 samples from different individuals obtained from the Federal Research Center of Fundamental and Translational Medicine, from 1 April to 30 May 2020 in Novosibirsk region (population 2.8 million people). We identified 6.49% positive for SARS-CoV-2 cases out of the total number of diagnostic tests, and 42% of them were from asymptomatic people. We also detected two asymptomatic people, who had no confirmed contact with patients with COVID-19. The highest percentage of positive samples was observed in the 80+ group (16.3%), while among the children and adults it did not exceed 8%. Among all the people tested, 2423 came from a total of 80 different destinations and only 27 of them were positive for SARS-CoV-2. Out of all the positive samples, 15 were taken for SARS-CoV-2 sequencing. According to the analysis of the genome sequences, the SARS-CoV-2 variants isolated in the Novosibirsk region at the beginning of the pandemic belonged to three phylogenetic lineages according to the Pangolin classification: B.1, B.1.1, and B.1.1.129. All Novosibirsk isolates contained the D614G substitution in the Spike protein, two isolates werecharacterized by an additional M153T mutation, and one isolate wascharacterized by the L5F mutation.

## 1. Introduction

A novel severe acute respiratory syndrome coronavirus SARS-CoV-2, which belongs to the group betacoronavirus and causes the respiratory disease COVID-19, was declared a pandemic by WHO on 11 March 2020 [1,2]. The first outbreak was reported from Wuhan, China, in December 2019, later the virus spread across the world. Since20 April 2022, over 506 million cases have been reported worldwide and more than 6.2 million deaths have been confirmed, making the COVID-19 pandemic one of the deadliest in history [2,3]. By the beginning of 2020, SARS-CoV-2 had already shown itself to be a dangerous causative agent of acute respiratory viral infections with an unpredictable course and a high mortality, affecting various organs through direct infection or through the body’s immune response [2,3,4]. The virus is spread via contact with infected individuals by airborne droplets through inhalation, with the virus sprayed in the air when coughing, sneezing, or talking, as well as through touching the mucous membranes with hands that had touched surfaces with the virus on them [2,4,5,6]. Due to the sharp increase in the number of cases requiring diagnosis and hospitalization, the burden on the healthcare systems in all countries of the world has also increased. Transmissions in healthcare settings, both nosocomial and to healthcare personnel, were first reported in the early period of the Wuhan outbreak [1,2,4]. In most cases, the infectious status of the patient was unknown and transmission was associated with emergency procedures such as intubation [5,6,7].

The description of the epidemiological processes from the beginning of a pandemic is of fundamental importance, both for epidemiology and virology, and for genetics and the social sciences. The SARS-CoV-2 epidemic is the first epidemic in the history of humankind to be traced from the beginning in such detail [8]. Investigating the variations and characteristics of the SARS-CoV-2 variants might help to assess the risks and develop better treatment and prevention strategies [9]. The development of pandemics is also studied in detail at the level of individual cities, at the level of countries, and the world.

The outbreak in Russia started in Moscow and Saint Petersburg and later spread to the remote regions [10], such as the Novosibirsk region (a federal subject of Russia, which is located in Siberia with a population of 2.8 million). Its administrative and economic center is the city of Novosibirsk, with a population of 1.6 million making it the third most populous city in Russia. The city is a major commercial, industrial, and transport hub.It is served by the Trans-Siberian Railway and the Tolmachevo Airport, which together connect Novosibirsk with Russia’s cities and most of the countries of Europe and Asia. The first case of the disease in the Novosibirsk region was detected on 19 March 2020 in a 47-year-old resident of Novosibirsk, who arrived in the city from Italy in transit through Moscow [11].

In our work, we used both the incidence data and phylogenetic approaches combined with evolutionary, demographic, and epidemiological concepts which have helped to track the virus genetic changes, to identify emerging variants, and to inform the public health strategy [8,9].

## 2. Materials and Methods

### 2.1. Collecting Samples

All aspects of the study were approved by the Ethics Committee of the Federal State Budget Scientific Institution: the“Federal Research Center of Fundamental and Translational Medicine”. Written informed consents were obtained from all the tested people or their parent or official representatives prior to sample taking.

At the initial stage of the COVID-19 pandemic in the Russian Federation, in April–May 2020 in the Novosibirsk region, 13,699 samples of biological material were collected as part of the work on the mass diagnosis of a new coronavirus infection. Biological material for the detection of SARS-CoV-2 (from nasopharyngeal swabs) was received from the residents of the Novosibirsk region who showed symptoms of acute respiratory illness andwho had had contact with patients with a confirmed diagnosis of COVID-19; from people who crossed the Russian border; and from healthcare professionals. For the epidemiological study and for the analysis of age- or sex-related morbidity, the following data were collected: the date of sample collection, the sex and age of the person from whom the biological material was taken, the presence or absence of symptoms, and the reason for taking the sample (contact with people with a confirmed diagnosis of COVID-19, crossing borders of the Russian Federation, arrival in Novosibirsk from another city of the Russian Federation, etc.).

### 2.2. Detection of SARS-CoV-2 RNA

All the samples were tested with the SARS-CoV-2 test systems (RealBest RNA SARS-CoV-2) according to the manufacturer’s (Vector-Best, Novosibirsk, Russia) methodology. The presence of SARS-CoV-2 RNA in the samples was confirmed by a real-time RT-PCR test and analyzed for epidemiological and genomic features. Only reliable results were included in the study. We have also excluded data relating to the same person, in order to calculate the number of people who have been positive/negative for SARS-CoV-2 RNA, instead of the number of positive/negative tests. In the case of a positive and a negative result, only a positive record was included into the study; in the case of several positive results from the same person, only the first positive result was included. There were no cases of re-infection (a sequence of positive, negative, and again positive analysis) during the entire observation period of 9 weeks.

### 2.3. SARS-CoV-2 Genomes

Of all the positive samples, 15 samples were taken for SARS-CoV-2 sequencing. The samples for sequencing were selected from several healthcare institutions located in the different districts of the Novosibirsk region. Complete genome NGS sequencing was performed using the Illumina MiSeq platform and the associated reagent kits, also from Illumina, according to the manufacturer’s methodology. RNA wasextracted using theQIAamp Viral RNA Mini Kit. Whole-genome amplification was performed using the ARTIC-protocol. DNA libraries were prepared using a Nextera DNA Flex Library Prep kit (Illumina, San Diego, CA, USA). Sequencingof the DNA libraries was conducted with a reagent kit, version 3 (600-cycle), on a MiSeq genome sequencer (Illumina). The consensus sequences were generated using Bowtie software. For genome-wide sequences, BLAST analysis was performed. In order to build phylogenetic dendrograms, the nucleotide sequences determined by sequencing, the closely related sequences determined by BLAST analysis, and the reference sequences belonging to the main genetic groups of SARS-CoV-2, were included in the general multiple alignment. The multiple alignment was performed using the MUSCLE [12], and its editing, including translation of the nucleotide sequences to amino acid sequences, was performed using the BioEdit software. Phylogenetic analyses were performed with MrBayes 3.2.7 utilizing a generalized time-reversible substitution model with a gamma-distributed rate variation across sites and a proportion of invariable sites (“GTR+I+Γ”). The phylogenetic tree was visualized using FigTree, version 1.4.4.

## 3. Results

### 3.1. Epidemiological Analysis

A total of 13,699 samples from different individuals was analyzed. The ages of the tested people and the number of positive cases in each age group is shown in Table 1. The division by age reflects the structure of society: preschoolers, schoolchildren, adults(19–60 years), the retired, and the elderly. There were 889 (6.49%) positive samples identified and SARS-CoV-2detectionsignificantly depended on age (χ^2^ = 104.667, *p* < 0.001).

In addition to determining the absolute number of positive tests, we also calculated the percentage of positive samples from the total number of tests done and the percentage of positive samples from each age group (0–9 years, 10–19 years, 20–29 years, 30–39 years, 40–49 years, 50–59 years, 60–69 years, 70–79 years, and 80+ years). Here we present a second version of the division into age groups for more detailed information. Among all the positive cases, there were few elderly people (80+ years–6.6%) (Figure 1a). However, when looking at the incidence within each age group, the retirees and the elderly people were more likely to have positive tests than the working adults (χ^2^ = 69.337, *p* < 0.001). The highest percentage of positive samples was observed in the elderly people (60+) with the maximum in the 80+ age group (16.3%), while among children and adults it did not exceed 8% (Figure 1b). The detection of the virus in the children did not significantly differ from the incidence in the adults. A similar age pattern was observed at the beginning of the pandemic in Italy where the most affected people were in nursing homes [1,2,13].

Despite the statements that children did not get sick in the first wave of COVID-19, about 4 percent of all cases were in children and 5% were in teenagers (Figure 1a), which contributed to the spread of infection through schools and children’s groups to families and older relatives. However, the incidence within the children in the 0–9 years age group was 4.3%, which is slightly lower than the incidence in the adults and teenagers (10–19 years—6.6%, more than 5 % in the other age groups, see Figure 1b).

Only half of the positive samples were obtained from symptomatic people and 42% were asymptomatic carriers of the disease (Figure 2). We detected fiveasymptomatic people, re-tested after recovery from COVID-19, but positive for SARS-CoV-2 RNA. Cases of long-term positivity to SARS-CoV-2 after recovery from COVID-19 are described in the literature [14]. We also detected two asymptomatic people, who had no confirmed contact with patients with COVID-19. It is possible that some of these people could develop symptoms later, but there were those who did not develop symptoms of COVID-19 during the observation period. Asymptomatic (never developing symptoms) and pre-symptomatic (testing positive prior to symptom development) carriage of SARS-CoV-2 has been confirmed in several studies [2,7,15,16,17]. Of the total number of positive samples 61% were obtained from females.

Among all the people tested, 2423 came from different countries and 934 came from other Russian cities of 80 different destinations. For 321 identified cases of infection, it was known that there was contact with a carrier of SARS-CoV-2 and the rest were tested in accordance with the Regulation. Among all the positive cases, 27 people came to the Novosibirsk region from other regions and countries: 16 were from Moscow, 4 from Saint Petersburg, 3 from Yakutia, 3 from Thailand and 1 from Spain. According to the official data, the first case of the disease in the Novosibirsk region was detected on19March 2020 in a person who arrived from Italy in transit through Moscow [11]. Our laboratory began mass testing on1 of April 2020; however, there were only a few cases of SARS-CoV-2 detected at the beginning of April in Novosibirsk. We analyzed the number of positive cases for nineweeks in total fromApril 1 to 30 May 2020. The incidence of positive cases increased unevenly during the first eightweeks, showing a slight decline at four and fiveweeks, followed by a sharp increase between six and eight weeks. By the end of May (week nine), other laboratories inNovosibirsk had joined the testing;therefore, only partial data was available to us. Here we show the number of positive cases expressed as a percentage of the total number of tested people per week (Figure 3).

### 3.2. The Genetic Diversity of SARS-CoV-2 in the Novosibirsk Region at the Beginning of the Pandemic

Notably, out of 15 sequenced SARS-CoV-2 sequences, 10 were taken from people with ARI symptoms such as fever, coughing, sore throat, and pneumonia who applied to hospitals at the beginning of the first wave. Five sequences were isolated from people who were not experiencing symptoms of COVID-19 at the time of the collection of biological material (three had contact with people infected with SARS-CoV-2).

The dates of collection of the samples of biological material, as well as the sex and the age of the tested people are presented below in Table 2.

According to the analysis of the genome sequences of the studied samples, the SARS-CoV-2 sequences detected at the beginning of the pandemic (from 14 Aprilto 7 May, i.e., within 24 days) belonged to three (G, GR, and GH) phylogenetic clades according to the classification GISAIDand three clades according to the Pango(Phylogenetic Assignment of Named Global Outbreak Lineages, PANGOLIN) nomenclature: B.1, B.1.1, and B.1.1.129. At the same time, seven out of fifteen samples belonged to the clade G (B.1); five viruses belonged to the clade GR (B.1.1); one strain belonged to the clade GR (B.1.1.129); and two belonged to the clade GH (B1). Thus, the GISAID clade GR is represented by two clades according to the Pango classification (B.1.1 and B.1.1.129) and the Pango clade B.1 covers two GISAID clades: G and GH.

The hCoV-19/Russia/Novosibirsk-RII27317S/2020 isolated on 30 April 2021, according to the Pango classification, belonged to the B.1.1.129 genetic line, also called the Russian Lineage. At present, only 70 genome sequences of this line are known [18]. Most of them have been identified within the territory of the Russian Federation, two in Germany, and one in Mexico. According to GISAID the first version of this line was discovered on 4 September 2020 in St. Petersburg.

Phylogenetically, the seven Novosibirsk sequences of clade G (Pango-B.1) form a separate cluster, which is distant from the genetic variants of this clade and included in the dendrogram, as a result of the BLAST analysis (Figure 4). On the other hand, the Novosibirsk sequences of clade B.1.1–GR did not form a single cluster, but rather formedthree separate groups. The hCoV-19/Russia/Novosibirsk-RII27321S/2020 was most phylogenetically related to the variants of clade G (B.1) and GR (B.1). The viruses hCoV-19/Russia/Novosibirsk-RII27313S/2020 and hCoV-19/Russia/Novosibirsk-5884/2020 are similar to each other and are closely related to the variants of SARS-CoV from another city in the Russian Federation, known as Irkutsk. The hCoV-19/Russia/Novosibirsk-3886/2020 and the hCoV-19/Russia/Novosibirsk-RII27322S/2020 are similar to each other and form a generic phylogenetic cluster with the SARS-CoV-2 variants from Europe (Latvia, Germany, Netherlands, and RF) and East Asia (South Korea and Hong Kong). The Novosibirsk viruses of clade B.1–GH hCoV-19/Russia/Novosibirsk-RII27316S/2020 and hCoV-19/Russia/Novosibirsk-RII27320S/2020 form a subcluster in a phylogenetic cluster with the SARS-CoV-2 variants from the USA, Australia, Canada, Colombia, and France. The hCoV-19/Russia/Novosibirsk-RII27317S/2020 of clade B.1.1.129–GR is phylogenetically distant from most of the Novosibirsk viruses of clade B.1.1–GR and is related to sequences from Saint-Petersburg (Russia, B.1.1.129) and from Germany (the B.1.1 and B.1.1.385 lineages).

To assess the heterogeneity of the pool of primary structures of the genomes of the studied viruses, including that within the individual clades and genetic lines, a heat map based on the matrix of pairwise genetic distances was built (Figure 5). According to the heat map, within the GISAID clade, the G and GH sequences were minimally separated from each other, while the isolates of the GR clade were more different from each other. In addition, of all the three clades, the GR and the GH were the most distant from each other.

To assess the variability of the SARS-CoV-2 proteins, an analysis of the non-synonymous nucleotide substitutions in the corresponding coding regions of the virus genome was carried out. As a result, after translation of the sequences, the amino acid substitutions were identified in the primary structures of the SARS-CoV-2 proteins. It was found that the amino acid substitutions occurred in 11 out of 25 proteins of the virus (Figure 6). The main antigen of SARS-CoV-2 is the Spike protein (S). All the Novosibirsk isolates contained the D614G substitution in the Spike protein, two isolates were characterized by an additional M153T mutation, and one isolate was characterized by the L5F mutation. Most of the amino acid substitutions were localized in the other 10 proteins and probably did not affect the change in the antigenic characteristics of SARS-CoV-2. Thus, 12 out of 15 of the Novosibirsk isolates contained identical main antigens—Spike proteins. The mutation D614G, according to the literature, promotes an increase in virus replication and increases the infectivity of SARS-CoV-2 both in cell cultures and in animal models.

Thus, at the initial stage of the pandemic in April–May, genetically different variants of SARS-CoV-2 belonging to several phylogenetic groups circulated in Novosibirsk. Phylogenetically different variants of SARS-CoV-2 were isolated from samples of biological material collected from patients of the same hospital over two days: hCoV-19/Russia/Novosibirsk-5897/2020, hCoV-19/Russia/Novosibirsk -5895/2020, hCoV-19/Russia/Novosibirsk-5890/2020 clade B.1–G, and hCoV-19/Russia/Novosibirsk-5884/2020 clade B.1.1–GR. In addition, seven phylogenetically related SARS-CoV-2 clade B.1–G variants were isolated from patients in three different hospitals.

## 4. Discussion

Phylogeographic analysis is used to track the circulation of SARS-CoV-2 using the ratio of genomic sequences of viruses in relation to the information about the location of sampling [8]. Phylogenetics combined with epidemiology provides a better understanding of the epidemiological processes that occur during the spread of viral infections. The lack of available SARS-CoV-2 genomes from certain areas may reduce the likelihood that these areas will be identified as a geographical source of a particular line/clade. By the time the introduction into Russia had started, the virus had already spread through other countries with the same variant frequently present at multiple locations [1,10,11]. Phylogenetic analysis indicates that the Russian samples represent much of the global diversity of the SARS-CoV-2 evolutionary tree [10]. The first cases of the detection of SARS-CoV-2 were registered in Moscow and St. Petersburg and then spread to the distant regions. Most samples corresponded to those wide-spread in Europe such as the B.1, B.1.1, and B.1.129 lineages, while the predominantly Asian A, B, and B.2 lineages were rare (Pango nomenclature) [10].

We discovered that the sequences related to the first wave of the epidemic in Novosibirsk and isolated in April, also belonged to these lineages B.1, B.1.1, and B.1.129, which indicates that the infection was caused not only by a variant of the coronavirus which is close in its genetic and molecular-biological characteristics to the primary, so-called “Wuhan” variant, but also to the European variants. At the beginning of the first wave in Novosibirsk we tested 2423 people, who had arrived in Novosibirsk from 80 different destinations, for SARS-CoV-2 and only 27 of them were positive. According to our data, the importations of SARS-CoV-2 into Novosibirsk were not from China, but from Moscow, St. Petersburg, Yakutia (the northern region of Russia), as well as from Thailand and Spain. Similar results have been obtained for other Russian regions [10,11,19,20,21,22].

According to world data [23], the first epidemic wave did not proceed in the same way in the different countries and cities [1,2,8,10,13,19,20,21,22,23,24,25,26,27]. The number of cases in China, Korea, Germany, and Russia was highest in the age group between 20 and 59 years and smallest in children (aged 0–9 years). In Italy, the majority of the cases in the first wave were in the 80+ age group, as the infection spread through nursing homes [13,19]. In Russia, the spread of the infection was by travelers in the beginning and later within their families [10].

The incidence structure in the first wave was also similar in Novosibirsk, Moscow [21], and Lipetsk [22]. The largest number of cases corresponded to the largest group, however, when normalized by the number of tests, it is revealed that the incidence in the group of 65 years and older exceeded the incidence rate in the other population groups. When normalized per 100 thousand population, A. B. Komissarov et al. showed that the incidence of children was also almost 2–3 times lower than the incidence of adults [7]. According to https://rosstat.gov.ru/ (access date 1 September 2022) in Novosibirsk region there are 232,093 children under 6 years old, 1,504,246 adults aged 20–60, and 99,715 adults aged 80+ [24].Since, there are fewer children than adults in total, and there were very few tests done in the beginning of the pandemic, the normalization by the number of tests in each age group is a more accurate way to estimate the incidence ineach age group. With this approach we find out that the children wereonly slightly less ill than the adults. The gender structure of the diseased was shifted towards women both in Moscow [8], Lipetsk [10], and in Novosibirsk, suggesting that women weremore likely than men to seek preventive care [28]. The gender structure for the Novosibirsk region is 1.15 female per 1 male person [29]. According to worldwide data, men were more affected than women, they tolerated the disease more severely, and they weremore likely to have symptoms [30]. In addition, here we have had an opportunity to test not only symptomatic, but also asymptomatic people and revealed a gender shift of the diseased towards women.

It is especially important that at the beginning of the epidemic, there was an active spread of SARS-CoV-2 by asymptomatic carriers of infection. At the beginning of the first wave, it was believed that asymptomatic carriage of SARS-CoV-2 was impossible and only symptomatic patients were contagious.We also detected two asymptomatic people, who had no confirmed contacts with patients with COVID-19,but were positive for SARS-CoV-2 RNA. Asymptomatic carriage of SARS-CoV-2has been confirmed in several studies [4,10,11,12]. Only the extensive testing of contact people and visitors from epidemiologically disadvantaged regions revealed that about half of the identified cases of infection were asymptomatic. The official aggregated incidence data cannot determine the true number of asymptomatic carriers, and data from hospitals only contain information about patients with symptoms, so our study, consideringthe personal data on contacts, travel, and symptoms, is especially relevant. We did not detect cases of re-infection (a sequence of positive, negative, and again positive analysis) during the entire observation period, but later such cases began to be recorded in many countries. The absence of such cases at the beginning of the pandemic is explained by the short period of observation, since the antibodies that were developed during the first disease continue to protect for some time.In addition, at the beginning of the pandemic, only a small number of strains were circulatingand they did not differ significantly from each other; while later such a large number of differences had accumulated that the effectiveness of the immunological protection decreased. We revealed that all ages of the population are susceptible to SARS-CoV-2 infection and this is in good agreement with world data [31]. Since we tested people with symptoms of acute respiratory illness, people who had contact with patients with a confirmed diagnosis of COVID-19, people who crossed the Russian border, and healthcare professionals, the sample structure is heterogeneous. It is likely that most elderly people had symptoms or contacts with COVID-19 patients, while adults (19–60 years) may have been both symptomatic and have been travelers or healthcare workers.

## 5. Conclusions

The first wave of the new coronavirus SARS-CoV-2 infection, which officially began in the Novosibirsk region at the end of March 2020, is characterized by a rapid spread among all population groups. The lowest incidence was among preschool children, and the incidence in schoolchildren and adults was at the level of 5–7%of the total number of tested people of the corresponding age group. The retired and especially the elderly people were more susceptible to SARS-CoV-2. We found that asymptomatic carriage (42%) contributes significantly to the spread of COVID-19, and that children get sick only slightly less often than adults. The highest initial incidence was in the age group over 80 years and most of the positive samples were obtained from female people, which is in line with the global trend.

We identified multiple origins of SARS-CoV-2 in Novosibirsk mainly from Moscow and St. Petersburg as well as from Thailand and Spain. Of the 15 sequenced samples, 7 corresponded to the B.1 (G) lineage, 5 belonged to B.1.1 (GR), 2 belonged to the B1 (GH) and 1 belonged to the genetic line B.1.1.129 (GR), also called the Russian lineage (according to Pango, https://cov-lineages.org, access date 09/01/2022).

## Figures and Tables

**Figure 1 viruses-14-02036-f001:**
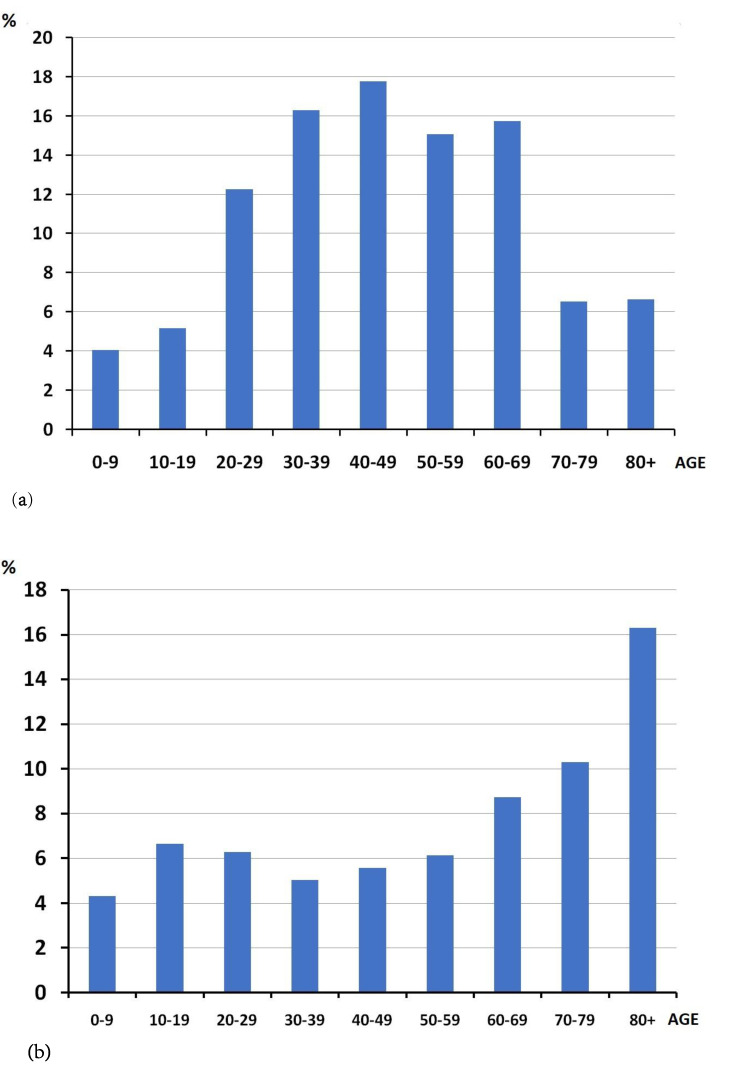
The percentage of positive cases per age group: (**a**) expressed as a percentage of the total number of positive cases; (**b**) expressed as a percentage of the total number of tested people of each corresponding age group. (**a**) the percentage of positive cases in each age group of the total number of positive cases. (**b**) the percentage of positive cases in each age group of the total number of tested people of each corresponding age group.

**Figure 2 viruses-14-02036-f002:**
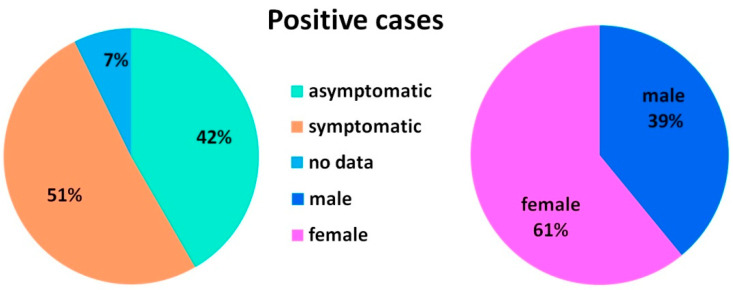
The percentage of symptomatic and asymptomatic people (**left**) and the percentage of male and female (**right**) among the positive for SARS-CoV-2 RNA.

**Figure 3 viruses-14-02036-f003:**
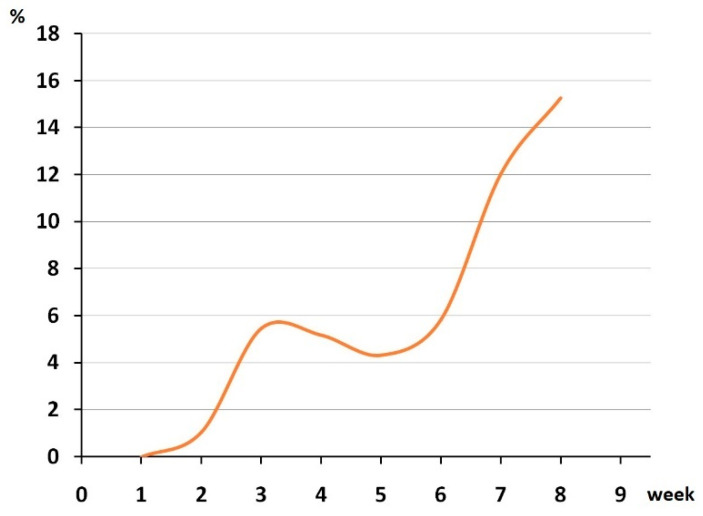
The number of positive cases, expressed as a percentage of the total number of tested people per week.

**Figure 4 viruses-14-02036-f004:**
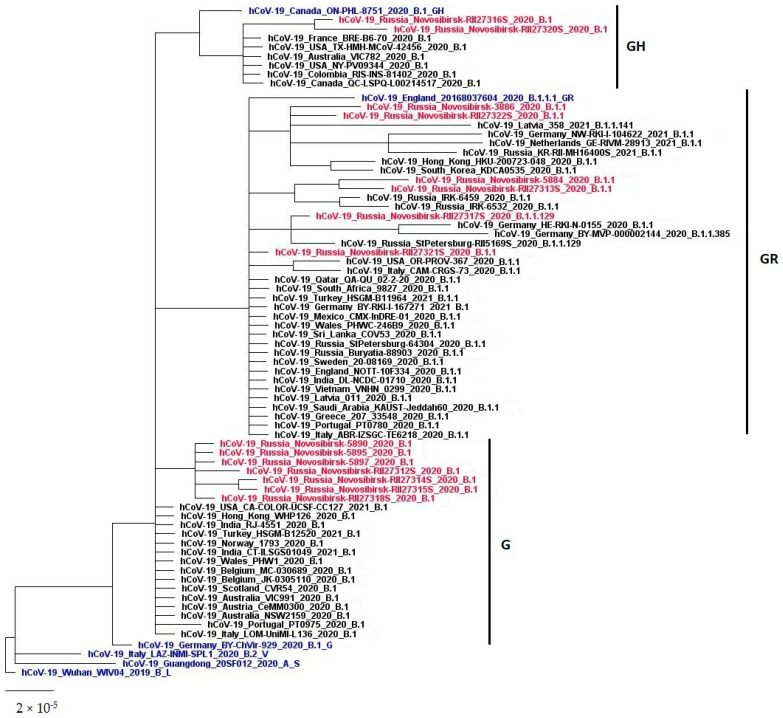
The phylogenetic dendrogram performed with MrBayes 3.2.7 and based on the whole genome nucleotide sequences of SARS-CoV-2. Red label—sequences of Novosibirsk isolates; blue label—reference sequences (according to GISAID EpiCoV).

**Figure 5 viruses-14-02036-f005:**
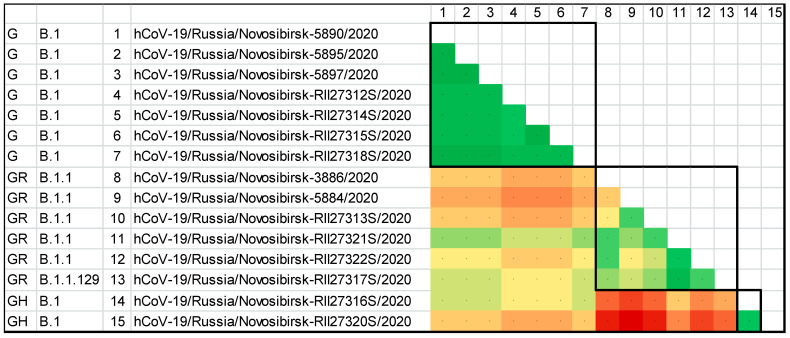
The heat map of the genetic distances between the genomes of SARS-CoV-2 isolates from Novosibirsk. The color gradient from green to red corresponds to an increase in pairwise genetic distances.

**Figure 6 viruses-14-02036-f006:**
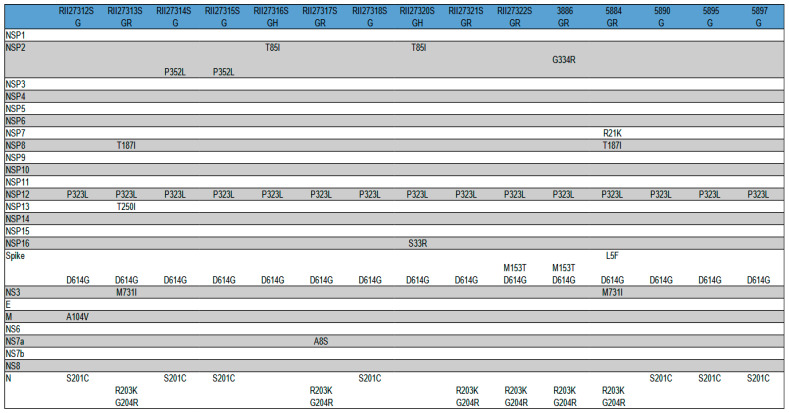
Amino acid substitutions in SARS-CoV-2 proteins from Novosibirsk according to the Wuhan reference hCoV-19/Wuhan/WIV04/2019 (WIV04).

**Table 1 viruses-14-02036-t001:** The number of tests and positive cases in the different age groups.

Age (Years)	Number of Tests	Positive
0–6	544	21
0–9	833	36
10–19	693	46
20–29	1738	109
30–39	2884	145
40–49	2833	158
50–59	2189	134
60–69	1604	140
70–79	563	58
80+	362	59
Total	13,699	889

**Table 2 viruses-14-02036-t002:** The genetic and epidemiological features of sequenced samples.

	Name	Pango	GISAID	Date	Sex	Age	Symptoms	Contact with an Infected Person
1	hCoV-19/Russia/Novosibirsk-3886/2020	B.1.1	GR	14 April 2020	Female	47	Yes	
2	hCoV-19/Russia/Novosibirsk-5897/2020	B.1	G	25 April 2020	Male	77	Yes	
3	hCoV-19/Russia/Novosibirsk-5895/2020	B.1	G	25 April 2020	Female	57	Yes	
4	hCoV-19/Russia/Novosibirsk-5890/2020	B.1	G	25 April 2020	Female	64	Yes	
5	hCoV-19/Russia/Novosibirsk-5884/2020	B.1.1	GR	26 April 2020	Female	46	Yes	
6	hCoV-19/Russia/Novosibirsk-RII27318S/2020	B.1	G	29 April 2020	Female	59	No	Yes
7	hCoV-19/Russia/Novosibirsk-RII27315S/2020	B.1	G	29 April 2020	Male	53	Yes	
8	hCoV-19/Russia/Novosibirsk-RII27314S/2020	B.1	G	29 April 2020	Female	47	Yes	
9	hCoV-19/Russia/Novosibirsk-RII27313S/2020	B.1.1	GR	29 April 2020	Female	68	No	Yes
10	hCoV-19/Russia/Novosibirsk-RII27312S/2020	B.1	G	29 April 2020	Female	70	No	No ^1^
11	hCoV-19/Russia/Novosibirsk-RII27317S/2020	B.1.1.129	GR	30 April 2020	Female	66	Yes	
12	hCoV-19/Russia/Novosibirsk-RII27316S/2020	B.1	GH	4 April 2020	Male	47	Yes	
13	hCoV-19/Russia/Novosibirsk-RII27322S/2020	B.1.1	GR	7 April 2020	Female	41	Yes	
14	hCoV-19/Russia/Novosibirsk-RII27321S/2020	B.1.1	GR	7 May 2020	Female	60	No	Yes
15	hCoV-19/Russia/Novosibirsk-RII27320S/2020	B.1	GH	7 May 2020	Female	47	No	No ^1^

^1^ Two strains were isolated as a result of the detection of asymptomatic cases without contact with an infected person.

## Data Availability

The sequence data presented in this study are openly available in GISAID database.

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
