# Peer review of "Genomic and Epidemiological Features of COVID-19 in the Novosibirsk Region during the Beginning of the Pandemic"

_viruses, 2022, doi:10.3390/v14092036_

Round 1

Reviewer 1 Report

The manuscript of Palyanova and colleagues describes the first wave of the Covid-19 pandemic (April 1 to May 30, 2020) in the Novosibirsk region. This might be of interest for the scientific community. I suggest to shorten the discussion and to focus more one the own data and its discussion. Given epidemiological data needs to be completed and presentation needs to be reworked.

Although I don't feel qualified to properly judge about the English, language and style should be reworked, some sentences read bit odd. Maybe a native speaker might rework the manuscript.

Specific comments:

Lines 114f, please include more details for the tree reconstruction method, e.g. the evolutionary model applied; I suggest to additionally use a MrBayes method

Line 119, replace “sample” by “person” or “patient”

Table 1, why is there another definition of age groups used as in Fig 1? (confusing), another column with disease outcome (“% recovered” or something) would be fine

Fig 1, the labeling of the axis is missing

Fig 1, the difference between both charts might simply mean a higher test rate for the groups of older people? (how were the test numbers in different age groups?); maybe adapt lines 332-334

Lines 149f, does this mean persistent infection? Where that immunosuppressed persons?

Fig 2, the labeling of the axis is missing

Lines 179, 181, exchange “variants” by “sequences”

Table 2, column “contacts” is misleading without reading the text above the table (maybe: contact of asymptomatic cases)

Table 2, are the data in this table sufficiently anonymized?  

Line 190, the sentence is misleading; do you mean 70 genome sequences or GISAID entries?

Lines 203, 205, exchange “viruses” by “sequences”

Fig 4, maybe add Pangoline lineage and/or WHO label for orientation

Fig 4, legend, include the method used and explain what is meant with “reference sequences” (is it important to mark them here?)

Lines 240-242, for the results described here, no detailed methods are described; please add in the method section

Fig 6, legend, amino acid positions are given according to the Wuhan reference?

Discussion, the first two paragraphs are more or less redundant

Discussion, lines 295-298, this is redundant, compare lines 57-65

Discussion, lines 335-338, no data of the gender structure for the Novosibirsk region was shown in the results, please add; in addition, this seems to be in contrast to other data (worldwide, men were more affected) and needs to be discussed in detail

Line 367, what does “active adults” mean?

Minor points:

Lines 25 and 243, D614G (not 624)

Lines 32 and 265, unify the spelling of betacoronavirus

Lines 103-106, NGS method description might be done in more detail

Lines 130, 134, 139, add parentheses for the statement of Figures

Lines 147-149, adapt the order of described results in relation to Fig. 2 (or exchange both charts with each other)

Line 180, please explain “ARI” symptoms

Line 189, SARS-CoV-2

Line 196, B.1

Lines 229 and 242, bold letters for the Figure (as done in the entire ms)

Line 301, these lineages

Author Response

Dear Reviewer,

We are thankful for your comments on our article.

We appreciate all comments and tried to improve our manuscript according to them. We agree with your suggestions and a native speaker will make a grammar and sentence structure revision after the paper has been accepted (the company will provide an English Editing service, which is included in the APC).

  1. General comment: The manuscript of Palyanova and colleagues describes the first wave of the Covid-19 pandemic (April 1 to May 30, 2020) in the Novosibirsk region. This might be of interest for the scientific community. I suggest to shorten the discussion and to focus more one the own data and its discussion. Given epidemiological data needs to be completed and presentation needs to be reworked. Although I don't feel qualified to properly judge about the English, language and style should be reworked, some sentences read bit odd. Maybe a native speaker might rework the manuscript.

Response: Thank you for this valuable suggestion. We shorten and reworked the discussion, the epidemiological data was completed and the presentation was reworked.

Specific comments:

  1. Lines 114f, please include more details for the tree reconstruction method, e.g. the evolutionary model applied; I suggest to additionally use a MrBayes method

Response:  First of all we thank for the suggestion to use MrBayes for phylogenetic analysis. Phylogenetic analyses were performed with MrBayes 3.2.7 utilizing generalized time-reversible substitution model with gamma-distributed rate variation across sites and a proportion of invariable sites (”GTR+I+Γ”). The phylogenetic tree was visualized using FigTree, version 1.4.4.

Information was included to the “Materials and methods” section and to the legend for Fig.4.

  1. Line 119, replace “sample” by “person” or “patient”

Response:  Here “sample” means a group of tested persons or patients.  “The age structure of the sample” was replaced with “The age of tested persons”.

  1. Table 1, why is there another definition of age groups used as in Fig 1? (confusing), another column with disease outcome (“% recovered” or something) would be fine.

Response:  We presented two version of the division of age groups for more detailed information, to enable researchers of different fields (pediatrics, epidemiology) compare our results with other regions.

We have no information about disease outcome. Hospitals did not provide us with this information, and most of tested persons were not from hospitals, but from airplanes with travelers from abroad, their families, medical stuff and people with symptoms who were treated at home.

We modified the Table 1 according to your comments.

  1. Fig 1, the labeling of the axis is missing

Response:  Thank you for bringing this oversight to our attention. The labelings of the axis are added.

  1. Fig 1, the difference between both charts might simply mean a higher test rate for the groups of older people? (how were the test numbers in different age groups?); maybe adapt lines 332-334

Response:  We updated the legend to correctly describe the data shown:

Figure 1 (a): a percentage of positive cases in each age group of the total number of positive cases.

Figure 1 (b): a percentage of positive cases in each age group of the total number of tested persons of corresponding age group.

The test rate was very low in all age groups in the beginning of pandemic. The population of Novosibirsk region is 2.8 million people and we had only 13699 tests (0.0049 %) for that period. In Table 1 you can see how many tests were done for children 0-6 years (544) and for elderly 80+ (362). According to https://rosstat.gov.ru/ in Novosibirsk region there are 232,093 children under 6 years old, 1,504,246 adults aged 20-60 and 99,715 adults aged 80+. So there are fewer children than adults in total (Lines 332-334). We believe that at the beginning of the epidemic, when very few tests were done, normalization of the number of positive tests to the entire population does not reveal the true picture of the incidence. 

Also we adapted the Discussion: added more information about age and sex structure of population of Novosibirsk region.   

  1. Lines 149f, does this mean persistent infection? Where that immunosuppressed persons?

Response:  We have no information about immune status of tested persons, only the information that they indicated in the questionnaire. These five persons indicated their (a)symptomatic status as “no symptoms, recovered from COVID-19”.

Such cases are described in literature: D'Ardes D, Boccatonda A, Rossi I, Pontolillo M, Cocco G, Schiavone C, Santilli F, Guagnano MT, Bucci M, Cipollone F. Long-term Positivity to SARS-CoV-2: A Clinical Case of COVID-19 with Persistent Evidence of Infection. Eur J Case Rep Intern Med. 2020 May 11;7(6):001707. doi: 10.12890/2020_001707. PMID: 32523924; PMCID: PMC7279900.

  1. Fig 2, the labeling of the axis is missing

Response: There is no axis in Fig 2, but thanks to your comment we added axis labeling in Fig 3.

  1. Lines 179, 181, exchange “variants” by “sequences”

Response: We exchanged “variants” by “sequences”.

  1. Table 2, column “contacts” is misleading without reading the text above the table (maybe: contact of asymptomatic cases)

Response: Thank you for this comment, the column name was corrected to “contact with an infected person”.

  1. Table 2, are the data in this table sufficiently anonymized?  

Response: The data was sufficiently anonymized before sequencing, names and persons cannot be identified.

  1. Line 190, the sentence is misleading; do you mean 70 genome sequences or GISAID entries?

Response: Thank you for this comment. We changed ”viruses” to “genome sequences” and reference to B.1.1.129 Lineage Report was added.

B.1.1.129 Lineage Report. Karthik Gangavarapu, Alaa Abdel Latif, Julia Mullen, Manar Alkuzweny, Emory Hufbauer, Ginger Tsueng, Emily Haag, Mark Zeller, Christine M. Aceves, Karina Zaiets, Marco Cano, Jerry Zhou, Zhongchao Qian, Rachel Sattler, Nathaniel L Matteson, Joshua I. Levy, Raphael TC Lee, Lucas Freitas, Sebastian Maurer-Stroh, GISAID core and curation team, Marc A. Suchard, Chunlei Wu, Andrew I. Su, Kristian G. Andersen, Laura D. Hughes, and the Center for Viral Systems Biology. outbreak.info, (available at https://outbreak.info/situation-reports?pango=B.1.1.129). Accessed 31 August 2022

  1. Lines 203, 205, exchange “viruses” by “sequences”

Response: We changed “viruses” to “sequences”.

  1. Fig 4, maybe add Pangoline lineage and/or WHO label for orientation

Response: Thank you for this useful comment. The information about Pango-lineages was included to the sequences names (because some sequences belong to sublineages like B.1.1.129 and B.1.1.385) and GISAID clade classification was added as phylogenetic groups frames.

  1. Fig 4, legend, include the method used and explain what is meant with “reference sequences” (is it important to mark them here?)

Response: MrBayes utilizing GTR substitution model with gamma-distributed rate variation across sites and a proportion of invariable sites (”GTR+I+Γ”) was used for phylogenetic analysis. Reference sequences for the lineages and clades were given from GISAID EpiCoV (section«Full genome tree derived from all outbreak sequences»).

The legend for Fig 4 was changed to: Phylogenetic dendrogram performed with MrBayes 3.2.7 and  based on the whole genome nucleotide sequences of SARS-CoV-2. Red taxa - sequences of Novosibirsk isolates, blue taxa - reference sequences (according to GISAID EpiCoV).

  1. Lines 240-242, for the results described here, no detailed methods are described; please add in the method section

Response: Thank you for this comment. We added to the method section: “Multiple alignment was performed using the MUSCLE, and its editing (including translation of nucleotide sequences to amino acid sequences) was performed using the BioEdit software”.

  1. Fig 6, legend, amino acid positions are given according to the Wuhan reference?

Response: We corrected the legend to: “Amino acid substitutions in SARS-CoV-2 proteins from Novosibirsk according to the Wuhan reference hCoV-19/Wuhan/WIV04/2019 (WIV04)”.

  1. Discussion, the first two paragraphs are more or less redundant

Response: We are thankful for this important suggestion. We shorten and reworked the discussion, all redundant paragraphs were deleted.

  1. Discussion, lines 295-298, this is redundant, compare lines 57-65

Response: We shorten and reworked the discussion.

  1. Discussion, lines 335-338, no data of the gender structure for the Novosibirsk region was shown in the results, please add; in addition, this seems to be in contrast to other data (worldwide, men were more affected) and needs to be discussed in detail

Response: Thank you for this valuable suggestion. The epidemiological data was completed. We added more information about age and sex structure of population of Novosibirsk region. The discussion on gender was added. Here you can see a table of how men and women are affected in different countries. There is one more country with such a big shift towards women:

Russo C, Morello G, Malaguarnera R, Piro S, Furno DL, Malaguarnera L. Candidate genes of SARS-CoV-2 gender susceptibility. Sci Rep. 2021 Nov 9;11(1):21968. doi: 10.1038/s41598-021-01131-7. PMID: 34753980; PMCID: PMC8578384

  1. Line 367, what does “active adults” mean?

Response: “Active adults” mean adults 19 – 60 years old. We corrected “Active adults” to “adults (19 – 60)”.

Minor points:

Lines 25 and 243, D614G (not 624)

Response: Corrected to D614G

Lines 32 and 265, unify the spelling of betacoronavirus

Response: Corrected to betacoronavirus

Lines 103-106, NGS method description might be done in more detail

Response: NGS method description was added to Methods: Complete genome NGS sequencing was performed using the Illumina MiSeq platform and associated reagent kits, also from Illumina according to the manufacturer's methodology. RNA were extracted using QIAamp Viral RNA Mini Kit. Whole-genome amplification was performed using the ARTIC-protocol. DNA libraries were prepared using a Nextera DNA Flex Library Prep kit (Illumina). Suqencing of the DNA libraries was conducted with a reagent kit, version 3 (600-cycle), on a MiSeq genome sequencer (Illumina). The consensus sequences were genegated by using Bowtie software.

Lines 130, 134, 139, add parentheses for the statement of Figures

Response: parentheses were added

Lines 147-149, adapt the order of described results in relation to Fig. 2 (or exchange both charts with each other)

Response: The order of described results in relation to Fig. 2 was changed.

Line 180, please explain “ARI” symptoms

Response: The ARI symptoms are:  fever, coughing, sore throat, pneumonia

Line 189, SARS-CoV-2

Response: Corrected to SARS-CoV-2

Line 196, B.1

Response: Corrected to B.1

Lines 229 and 242, bold letters for the Figure (as done in the entire ms)

Response: Corrected to Figure

Line 301, these lineages

Response: Corrected to these lineages

Response: We thank you for all points. All of them were corrected according to your comments.

Kind regards,
Mrs. Natalia Palyanova

Reviewer 2 Report

Review Comments:

Overview: The authors analyzed the results of a single-center study of COVID incidence in Novosibirsk over a two-month period in 2020 and studied the genetic lineage of the COVID isolates in the population, from which they identified three SARS-CoV-2 lineages. The findings add to the understanding of SARS-CoV-2 origin and spread in the Novosibirsk population, and hence could be of significance in understanding the epidemiology and emergence of the virus in the area.

Major concerns: I do not have major concerns. The study analyzes retrospective data collected following the standard, IRB-approved protocol for the region, which the authors have explained at the end of the manuscript. 

I do think that the writing needs a careful review for grammar and sentence structure revision throughout the paper. Please see my comments below.

Minor concerns:

Sentence structure is odd at places throughout the text. Some, but not all, examples from the text, are listed below. Please conduct a careful review for grammar and sentence structure revision.

Examples:

In the abstract:

line #24: “…at the beginning of the pandemic are belonged to three phylogenetic lineages…” should be corrected as “…at the beginning of the pandemic belonged to three phylogenetic lineages…”

In the introduction section:

Line # 40-43: “The virus is spread via contact with infected individuals by air- borne droplets through the inhalation of droplets with the virus sprayed in the air when coughing, sneezing or talking, as well as through the virus getting on the surface with subsequent entry into the eyes, nose or mouth” - is confusing.

In the materials and methods section:

Lines #95-97: “In the case of a positive and negative result, only a positive record was left; in the case of several positive results of the same person, only the first positive result was left” - is confusing. By “result was left”, do you mean the results were excluded from analysis or were they included in the analysis? Please revise the sentence to remove the confusion.

In the discussion section:

Lines #264-266

Lines #272-273

In Figures:

Descriptions of titles for figure 1(a) and 1(b): The graphs are showing percentages, not the total numbers. Please revise the figure descriptions to correctly describe the data shown.

Figure 2 description: These charts are also showing percentages, not the total numbers. Please revise the figure descriptions to correctly describe the data shown.

Lines #142-143: “4,3%” should be corrected to “4.3%” and “6,6%” should be corrected to “6.6%”

Line #147: Please replace “female persons” with “females”

Other concerns:

The authors mention the sampled population to consist of “unique persons”. However, the characteristics that explain the uniqueness of the sampled subjects are not described clearly in the text.  Please include a clear explanation of what made the sampled subjects “unique”, in the materials and methods and the discussion sections.

In Materials and Methods section, line # 92: The authors describe that “Only reliable results were included in the study” – Please include an explanation of how the reliability was determined. 

Author Response

Dear Reviewer,

Thank you for your comments on our article.

We appreciate all comments and tried to improve our manuscript according to them.

  1. Major concerns:

I do not have major concerns. The study analyzes retrospective data collected following the standard, IRB-approved protocol for the region, which the authors have explained at the end of the manuscript. 

I do think that the writing needs a careful review for grammar and sentence structure revision throughout the paper. Please see my comments below.

Response: We agree with your suggestions and a native speaker will make a grammar and sentence structure revision after the paper has been accepted (the company will provide an English Editing service, which is included in the APC).

  1. Minor concerns:

Sentence structure is odd at places throughout the text. Some, but not all, examples from the text, are listed below. Please conduct a careful review for grammar and sentence structure revision.

Response: Thank you for this comments.

Examples:

  1. In the abstract:

line #24: “…at the beginning of the pandemic are belonged to three phylogenetic lineages…” should be corrected as “…at the beginning of the pandemic belonged to three phylogenetic lineages…”

Response: The line was corrected to: “…at the beginning of the pandemic belonged to three phylogenetic lineages…”

  1. In the introduction section:

Line # 40-43: “The virus is spread via contact with infected individuals by air- borne droplets through the inhalation of droplets with the virus sprayed in the air when coughing, sneezing or talking, as well as through the virus getting on the surface with subsequent entry into the eyes, nose or mouth” - is confusing.

Response: The line was corrected to: “The virus is spread via contact with infected individuals by airborne droplets through the inhalation of droplets with the virus sprayed in the air when coughing, sneezing or talking, as well as through touching mucous membranes with hands that touched surfaces with virus on them.”  

  1. In the materials and methods section:

Lines #95-97: “In the case of a positive and negative result, only a positive record was left; in the case of several positive results of the same person, only the first positive result was left” - is confusing. By “result was left”, do you mean the results were excluded from analysis or were they included in the analysis? Please revise the sentence to remove the confusion.

Response: The line was corrected to: “In the case of a positive and negative result, only a positive record was included into the study; in the case of several positive results of the same person, only the first positive result was included”.

  1. In the discussion section:

Lines #264-266

Lines #272-273

Response: We are thankful for this important suggestion. We shorten and reworked the discussion, all redundant paragraphs were deleted.

  1. In Figures:
  • Descriptions of titles for figure 1(a) and 1(b): The graphs are showing percentages, not the total numbers. Please revise the figure descriptions to correctly describe the data shown.

Response: We are thankful for this important suggestion. We corrected the figure descriptions:

Figure 1: The percentage of positive cases, per each age group.

Figure 1 (a): a percentage of positive cases in each age group of the total number of positive cases

Figure 1 (b): a percentage of positive cases in each age group of the total number of tested persons of corresponding age group

We also added axis labeling to Figure 1 and 3.

  • Figure 2 description: These charts are also showing percentages, not the total numbers. Please revise the figure descriptions to correctly describe the data shown.

Response: Thank you for this important comment. We corrected the figure descriptions to: “The percentage of symptomatic and asymptomatic persons (left) and the percentage of male and female (right) among positive for SARS-CoV-2 RNA”.

  • Lines #142-143: “4,3%” should be corrected to “4.3%” and “6,6%” should be corrected to “6.6%”

Response: Corrected to: “4.3%” and “6.6%”.

  • Line #147: Please replace “female persons” with “females”

Response: “female persons” was replaced with “females”. 

  1. Other concerns:
  • The authors mention the sampled population to consist of “unique persons”. However, the characteristics that explain the uniqueness of the sampled subjects are not described clearly in the text.  Please include a clear explanation of what made the sampled subjects “unique”, in the materials and methods and the discussion sections.

Response:  By “unique” we mean “without duplicates”: we excluded data relating to the same person, in case of multiple tests. So we consider unique, not repetitive persons in our work. We replaced the word “unique“ to “different individuals” in Line #15 and #118.

  • In Materials and Methods section, line # 92: The authors describe that “Only reliable results were included in the study” – Please include an explanation of how the reliability was determined. 

Response:  All the results were verified by the Russian Federal Service for Surveillance on Consumer Rights Protection and Human Wellbeing (Rospotrebnadzor) service according to mass diagnostic protocols.

Kind regards,
Mrs. Natalia Palyanova

Round 2

Reviewer 1 Report

no further comments